# Preparation of a New Type of Cemented Paste Backfill with an Alkali-Activated Silica Fume and Slag Composite Binder

**DOI:** 10.3390/ma13020372

**Published:** 2020-01-13

**Authors:** Qi Sun, Tianlong Li, Bing Liang

**Affiliations:** 1School of Civil Engineering, Liaoning Technical University, Fuxin, Liaoning Province 123000, China; sunqi@lntu.edu.cn (Q.S.); 471820438@stu.lntu.edu.cn (T.L.); 2School of Mechanics and Engineering, Liaoning Technical University, Fuxin, Liaoning Province 123000, China

**Keywords:** cemented paste backfill, alkali-activated silica fume and slag composite binder, strength formation mechanism, microstructure

## Abstract

A new type of cemented paste backfill (CPB) was prepared using sodium hydroxide (NaOH) as the activator, slag and silica fume (SF) as the binder, and tailings as the aggregate. The effects of proportion of replacement of 0%, 5%, 10%, 15%, and 20% silica fume on the properties of CPB were studied. The strength formation mechanism of CPB was explored through a combination of scanning electron microscopy (SEM), energy dispersive spectrometry (EDS), and Fourier transform infrared (FTIR) spectroscopy. The SEM images were analyzed by IMAGE J software, and the porosity of CPB with different silica fume contents was obtained. The results show that as the amount of silica fume increases, the unconfined compressive strength (UCS) increases first and then decreases. When the amount of silica fume was approximately 5%, CPB with a larger UCS can be obtained. When the silica fume content increased from 0% to 5%, because silica fume has good activity and small particles, more calcium silicate hydrate (C–S–H) gels and Mg-Al type layered double hydrotalcites (LDHs) were generated in CPB, which made it denser and improved its strength compared with the non-silica fume group. C–S–H gels were the main source of CPB strength. With a further increase in the amount of silica fume, thaumasite produced inside of CPB, reducing the content of C–S–H gels. Moreover, due to the expansion of thaumasite, it is easy to generate a large number of micro cracks in CPB, which weakens the strength of CPB.

## 1. Introduction

Tailings are solid waste formed by the beneficiation of metal minerals. According to statistics, the global average annual tailings output exceeds 10 billion tons and will continue to grow in the future [1,2]. The tailings discharged from a concentrator are often stacked into tailings dams. Accidents caused by dam breaks in tailings dams are common worldwide, and untreated tailings containing heavy metals that can easily contaminate land resources and groundwater resources, cause atmospheric pollution and endanger human health [3,4,5]. These issues have led many countries to undertake enormous fiscal expenditures to manage idle tailings [6]. Using tailings to prepare cemented paste backfill (CPB) and filling it into a mined-out area is not only conducive to controlling mining subsidence [7,8] but also avoids geological disasters and environmental pollution caused by tailings dam instability, making it a green and sustainable method [9,10]. As a by-product of the metallurgical industry, silica fume (SF) and slag have pozzolanic activity. Therefore, alkali activation is used to stimulate the activity of silica fume and slag as a composite binder, and tailings are used as an aggregate to prepare a new type of environmentally friendly CPB is of great significance.

In recent years, scholars have conducted in-depth research on the material composition and mechanical properties of CPB. Chen et al. [11,12] innovatively designed a large-scale similar stope model (SSM), tested the unconfined compressive strength (UCS) regularity of CPB at different locations, analyzed the mechanism, and concluded that the UCS reached a maximum at 60% of the SSM. CPB was prepared using phosphogypsum and building demolition waste to achieve clean production. Cao et al. [13,14,15] studied the effect of structural factors on the UCS of cemented tailings backfill (CTB), and considered the effects of different fiber additions on the strength and toughness of CTB. The UCS of CTB was higher with polypropylene fiber than with other fibers, and glass fiber had the best performance in terms of toughness. The change in the peak compressive strength of CTB at different loading rates was studied, showing a power function relationship. Qi et al. [16,17,18] proposed an intelligent method for predicting the UCS of CPB, constructed a constitutive model and a strength prediction model for CPB, and analyzed the hydration reaction mechanism of cement in CPB. Liu et al. [19,20,21] studied the early hydration heat, hydration mechanism, and kinetic parameters of CPB by isothermal calorimetry, focused on the effect of the tailings–cement ratio (TCR) on hydration heat, and established a prediction model for paste composition and rheological properties. A negative exponential relationship between the porosity of the CPB and UCS was also proposed. Zhang et al. [22] prepared a cement paste containing copper tailings (CPCT) and studied the effects of the copper tailings dosage and curing age on the porosity of the slurry; they concluded that the microstructure became increasingly compact with increasing age and copper tailings consumption. Wu et al. [23,24] prepared cemented coal gangue-fly ash backfill (CGFB) using coal gangue, fly ash (FA) and ordinary Portland cement (OPC) as constituent materials; they analyzed the effect mechanism of multi-field coupling on the performance of CGFB and concluded that increasing the OPC/FA ratio and curing temperature could accelerate the hydration reaction and obtain higher compressive strength. Zhao et al. [25] used Portland cement (PC) and FA as a binder to study the law of CPB strength development and local strain and noted that 4% PC + 3% FA can simultaneously meet the minimum thresholds and requirements for mining stability. Su et al. [26] prepared CPB from cement and lead-zinc tailings and used the dynamic, static, and toxic leaching method to control metal elements by adding 2 mg/L polyaluminium sulfate (PAS).

Some scholars have also studied the application of alkali-activated binder in CPB. Sun et al. [27] prepared a geopolymer cemented coal gangue-fly ash backfill (GCGFB) using alkali-activated fly ash as a binder. The fracture evolution and deformation of GCGFB were observed using the digital speckle correlation method (DSCM). He et al. [28] studied the use of lithium slag and FA to prepare a binder and mixed it with tailings to prepare cemented fine tailings backfill (CFTB); they obtained the optimal ratio of CFTB and analyzed the strength formation mechanism. Katpady et al. [29] prepared a geopolymer using shirasu as an aluminosilicate, sodium hydroxide and water glass as an alkali activator, and slag as an additive, and they concluded that the UCS and acid resistance of the specimens were significantly improved with the addition of slag compared with no slag addition. Feng et al. [30,31] used mechanical activation and the incorporation of calcium oxide to improve the pozzolanic activity of granular copper slag (GCS); they also prepared a binder, as well as a suitable binder for formulating CPB. Jiang et al. [32,33] studied the early strength and working characteristics of CPB using alkali-activated slag as a binder; they also studied the change in yield stress of CPB under different low temperatures and time conditions. Cihangir et al. [34,35] comparatively analyzed the mechanical properties of samples using full tailings (FT) and deslimed sulphide-rich tailings (DT) as aggregates and alkali slag as a binder and noted that samples with DT had better performance than those with FT. In addition, CPB with different types of binders and different amounts of binder was prepared using cement and alkali-activated blast furnace slag. It was concluded that the performance of alkali-activated blast furnace slag as the binder is superior to that of cement in both the short and long term. Qiu et al. [36] studied the effects of different amounts of FA and blast furnace slag (BFS) and the amount of alkali used on the properties of the FA/BFS base polymer; they concluded that after increasing the amount of slag and the amount of alkali, the compressive strength of the slag increased, but there was a certain loss in slump.

Some scholars have investigated the use of alkali-activated slag and silica fume to prepare concrete and mortar, Collins et al. [37] used condensed silica fume to partially replace slag in the alkali-activated slag binder system, found that the strength and the water demand were significantly increased. Rostami et al. [38] mixed the silica fume in alkali-activated slag concrete, studied the strength and impermeability of concrete under different silica fume proportions of replacement and the effects on concrete under different curing conditions. Ramezanianpour et al. [39] used NaOH, KOH, Na_2_SiO_3_, etc. as stimulants and used different doses of nanosilica and silica fume to replace slag to prepare mortar. Rashad et al. [40] used sodium silicate to activate slag/silica fume pastes, and compared its compressive strength and thermal shock resistance at different high temperatures.

These studies used a variety of solid wastes to prepare CPB and examined the CPB mechanical properties. The use of alkali-activated slag as a binder to prepare CPB has achieved certain results. Some scholars have replaced silica fume with slag in the alkali-activated slag system to prepare binder and applied it to the preparation of concrete or mortar. However, no research has been reported on the preparation of CPB with slag and silica fume as the binder. Silica fume has good activity and small particle size. The addition of silica fume to alkali-activated CPB may improve its performance; therefore, further research is necessary.

In this paper, alkali-activated slag and silica fume were used as binders and tailings were used as aggregates to prepare a new CPB to explore the influence of the proportion of slag and silica fume on the UCS of CPB. The evolution mechanism of CPB strength was analyzed by scanning electron microscopy (SEM), energy dispersive spectrometry (EDS), and Fourier transform infrared (FTIR) spectroscopy.

## 2. Materials and Methods

### 2.1. Materials

The main binder for this study was S95 fine slag powder produced by Shandong Kangjing New Material Technology Co., Ltd. China, and silica fume was produced by Luoyang Yumin Micro Silica Fume Co., Ltd. China. The specific surface area of the slag and silica fume was 397.6m^2^/kg and 414.3m^2^/kg. The aggregate is tailings taken from the Baoshan Iron Mine Concentrator in Anshan City, Liaoning Province, China. The particle size distribution of tailings and binders were characterized by BT-2003 laser particle size analyzer (Dandong Bettersize Instrument Co., Ltd., China) as shown in Figure 1. According to the Figure 1, tailings can be determined as fine aggregate. The maximum aggregate size was 398.2μm. The specific surface area of the tailings was 231.3m^2^/kg. The bulk density of the aggregate was 1752.2kg/m^3^. The chemical composition of the slag, silica fume, and tailings was determined using a Shimadzu XRF1800 spectrometer. The specific chemical compositions are shown in Table 1. The mineral composition of tailings was measured by X-ray diffraction (XRD). XRD tests were conducted using a D8 ADVANCE X-ray diffractometer. According to XRD pattern shown in Figure 2, the mineral components of tailings mainly include mineral components such as quartz, chlorite-serpentine, muscovite, and anorthite.

The alkaline catalyst used in this study was a commercially available solid sample of sodium hydroxide from Liaoning Quanrui Reagent Factory with a purity of 96%. Considering silica fume and slag as volcanic ash materials with high specific surface areas, to meet the mixture slump requirements and the CPB pumping requirements in the actual filling process, an appropriate amount of superplasticizer was used in the experiment to ensure that the slump was within the range of 150–200 mm. The superplasticizer was from Fuyang Chengnan Building Material Co., Ltd, China. It is a polycarboxylate-type superplasticizer in the form of a white powder, and the mixing water is tap water.

### 2.2. Test Methods

The mixture proportion is shown in Table 2. The proportion of silica fume in the binder was 0%, 5%, 10%, 15%, and 20%, corresponding to the SF0-SF20 group. After a large number of previous attempts, the experimental parameters were determined to be a water-binder ratio of 2.1 and an aggregate-binder ratio of 7, with an alkali content of 4% by binder mass (alkali content was 3.1% if calculated in terms of Na_2_O). Binder content was 9.86% of the total mass. The dosage of superplasticizer in each group is shown in Table 2. The mixing water used in the experiment consisted of two parts: one was used to dissolve the solid particles of NaOH and the superplasticizer, and the other was used to rinse the solution remaining in the glass beaker during the process of dumping the solution. To better dissolve the NaOH solid and superplasticizer, a certain amount of water was first added until the NaOH solid was fully mixed and dissolved. The superplasticizer was then added under continuous stirring until it was homogeneous, and the solution was left at room temperature for 10 min for further processing. After weighing the binder and aggregates used in the experiment, the aggregates were placed in a UJZ-15 mortar mixer manufactured by Shanghai Luda Experimental Instrument Co., Ltd., and stirred for 3 min. Then, the slag and silica fume were poured into the mixer in three portions and stirred for 3 min to ensure material homogeneity in the mixer. Finally, the pre-configured solution was poured into the mortar mixer and stirred well for 4 min. The slump test was performed immediately after the stirring was completed. The slump test results of SF0-SF20 group were: 195 mm, 180 mm, 170 mm, 175 mm, and 190 mm. The fresh mixture was poured into a cylindrical mold with a diameter of 50 mm and a height of 100 mm. Three test specimens were prepared at each age for UCS testing. All molds containing the mixture were placed on a shaking table and shaken for 40 seconds. The mold containing the mixture was sealed and placed in a concrete standard curing box in which the temperature was controlled at 20 (±1) °C, and the humidity was controlled at ≥96% for curing. Model of concrete standard curing box is YH-40B. After one day of curing, the CPB molds were demolded and keep the samples in the above environment. After 28 days of curing, uniaxial compression experiments, SEM, EDS, and FTIR experiments were performed. Uniaxial compression experiments were performed using a universal testing machine. A HITACHI S-3400N scanning electron microscope was used for SEM and EDS analysis at a voltage of 20.0 kV. A Nicolet iS50 Fourier infrared spectrometer was used for characterization.

## 3. Results and Discussion

### 3.1. Unconfined Compressive Strength

The UCS test results of specimens with different curing ages are shown in Figure 3. From the 3-day UCS, compared to the non-silica fume group, the increase in strength of the silica fume group was not significant because CPB had a slower reaction rate and fewer hydration products generated within 3 days of curing. After 7 days of curing, the UCS of each group began to grow rapidly because CPB had a certain early strength and reached its peak after 28 days. From the test results, it is clear that for the SF5 group, in which the slag and silica fume account for 95% and 5% of the binder, respectively, the UCS of each age was better than that of the other groups. Compared with the SF0 group, the SF5 group introduced a silicon source due to the addition of silica fume, and the slag in the mixture provided a large amount of calcium and aluminum. Under the action of the alkali-stimulated catalyst, a large amount of C–S–H gels and Mg–Al type LDHs were generated, which became the main source of CPB strength. These generated gels compress the pores, while at the same time, because the silica fume particles are very small, they have a good filling effect on the pores and make the microstructure denser. Therefore, the UCS was higher in the SF5 group than in the SF0 group. With an increase in the amount of silica fume, UCS showed a trend of increasing first and then decreasing, which is similar to the experimental results of silica fume mixed with concrete presented by Wetzel et al. [41] and Alanazi et al. [42]. It is worth noting that for the SF15 and SF20 groups, the UCS value of all curing ages was significantly lower than that of SF0, which indicates that the more silica fume that is used, the easier it is to decrease the UCS.

### 3.2. Microstructural Analysis

In this experiment, specimens with a curing age of 28 days were selected to observe their microstructure. The SEM micrograph of the hydration products of the SF0 group is shown in Figure 4a. From Figure 4a, a large amount of flocculent matter could be observed. According to the EDS spectrum of Figure 5, there were high contents of O, Mg, Al, Si, and Ca elements. It is presumed that the hydration products of slag are C–S–H and Mg–Al type LDHs, which are closely intertwined. This is similar to the findings of [43]. When slag contains MgO, C–S–H gel, and Mg–Al type LDHs (Mg_6_Al_2_CO_3_(OH)_16_·4(H_2_O)) are the main hydration products of slag. When they are tightly intertwined, O, Mg, Al, Si, and Ca elements coexist in a large amount in EDS. At the same time, white flocculent precipitates could also be observed. According to the EDS spectrum of Figure 6, there were high contents of O, Si, and Ca elements, while Na accounted for only 1.07%, and Al elements accounted for 3.08%. It was confirmed that the hydration product was the C–S–H gel. Many different-sized pores could be observed in the microstructure, which is one of the reasons why the compressive strength of this group is lower than that of SF5. An SEM micrograph of the hydration products of the SF5 group is shown in Figure 4b. Compared with the SF0 group, this group has a dense microstructure with few visible pores and very small pores. Analogous to the SEM image of the SF0 group, Mg-Al type LDHs and a white flocculent precipitate of C–S–H gels can also be observed, and these two hydration products were present in significantly higher amounts in the SF5 group than in the SF0 group. The calcium and aluminum elements contained in slag with the silicon elements contained in silica fume were better fused in SF5 than in SF0 to generate corresponding hydration products. The reaction equations [44] involved are shown below.
(1)$$SiO_{2}+Ca{\left(OH\right)}_{2}+H_{2}O\to CaO\cdot SiO_{2}\cdot xH_{2}O$$

An SEM image of the SF10 group is shown in Figure 4c. A small amount of fine needle-like hydration products can be observed. A small portion of these hydration products fills the pores, and most of the products adhere near the C–S–H gels. For the known hydration products in the concrete field, the fine needles are mainly ettringite and thaumasite, and their morphologies are very similar [45]. To discriminate whether this needle-like hydration product is ettringite or thaumasite, an analysis is shown in Figure 7. The EDS spectrum shows that 11.42% Si was present in the hydration product, and the other elements contained in thaumasite were reflected in the energy spectrum. It can be preliminarily speculated that the hydration product is thaumasite.

An SEM image of the SF15 group is shown in Figure 4d. The needle-like hydration products can be clearly observed, and layered substances appeared near these products. Haha et al. [43] indicated that the Mg-Al type LDHs (Mg_6_Al_2_CO_3_(OH)_16_·4(H_2_O)) is a substance with a layered structure, and C–S–H is covered with LDHs. Therefore, the EDS energy spectrum (Figure 8) contains a large number of O, Mg, Al, Si, and Ca elements. At the same time, it can be observed that the number of C–S–H gels compared with those in Figure 4c was significantly reduced.

An SEM image of the SF20 group is shown in Figure 4e. From the figure, many needle-like hydration products could be seen, and a very small amount of C–S–H gel was observed. This is probably due to the large amount of C–S–H gel consumed by the reaction and the thaumasite formed on the surface of the remaining C–S–H gel.

### 3.3. FTIR Spectral Analysis

To further determine the needle-like hydration products appearing in the SEM picture, this study performed an FTIR test on the SF20 group cured for 28 days. The FTIR spectrum is shown in Figure 9. According to a report by Ma et al. [46], the main wavenumbers for distinguishing thaumasite are 673 cm^−1^ for SiO_6_ stretching, 1100 cm^−1^ for ${SO}_{4}^{2-}$ stretching, 1400 cm^−1^ for ${CO}_{3}^{2-}$ stretching, and 1650 cm^−1^ and 3600–3200 cm^−1^ for H_2_O bending and stretching, respectively. It can be observed from the spectrum that a peak exists at a wavenumber of 669.43 cm^−1^. By comparing and analyzing the known wavenumber, it can be determined that the peak corresponds to the SiO_6_ stretching vibration. The silicon and oxygen in thaumasite have been shown to exist in the 6-coordinated form [47], and ettringite does not contain this chemical bond. The peak at 1074.52 cm^−1^ and the corresponding peak at 1100 cm^−1^ can be determined as ${SO}_{4}^{2-}$ stretching. The peak at 1426 cm^−1^ corresponds to the C–O bond stretching vibration, which is caused by the presence of ${CO}_{3}^{2-}$. The peaks at 1633.03 cm^−1^ and 3407.91 cm^−1^ can be determined as the bending and stretching of H_2_O, respectively. In contrast, a peak at 850 cm^−1^ for AlO_6_ is unique to ettringite but is not observed in the figure. Therefore, the needle-like hydration product can be determined as thaumasite. The main reason for the formation of thaumasite is that aluminate in slag reacts with the ferrite and sulfate in tailings to form ettringite, and calcium carbonate can form due to the reaction of calcium hydroxide with CO_2_ in solution or air. In this environment, the C–S–H gel reacts with ettringite, and the silicon element in the C–S–H gel continuously replaces the aluminum element in ettringite. With the progress of the reaction, ettringite is finally converted into thaumasite. In the SF5 group, no thaumasite was observed because the amount of silica fume was small, and this reaction was not sufficient to cause the above reaction. The reaction equation [48] involved is shown below.
(2)$$\begin{array}{l} Ca_{6}{\left[Al_{x}Fe_{\left(1-x\right)}{\left(OH\right)}_{6}\right]}_{2}{\left(SO_{4}\right)}_{3}\cdot 26H_{2}O+Ca_{3}Si_{2}O_{7}3H_{2}O+2CaCO_{3}+4H_{2}O\\ \to Ca_{6}{\left[Si{\left(OH\right)}_{6}\right]}_{2}{\left(CO_{3}\right)}_{2}{\left(SO_{4}\right)}_{2}24H_{2}O+CaSO_{4}2H_{2}O+2xAl{\left(OH\right)}_{3}\\ +2\left(1-x\right)Fe{\left(OH\right)}_{3}+4Ca{\left(OH\right)}_{2} \end{array}$$
(3)$$Ca{\left(OH\right)}_{2}+CO_{2}\to CaCO_{3}+H_{2}O$$

Thaumasite has been proven to be a swellable hydration product and easily cause micro-cracks in CPB [49]. With the increase of silica fume replacement proportion, the content of calcium element decrease, that is because the added silicon is not enough to produce more hydration products with the other elements, so the C–S–H gels generated during the same period are reduced. Meanwhile, C–S–H gels were the main source of CPB strength. Due to expansibility of thaumasite, a large number of micro-cracks were generated in the CPB, leading to a decrease in the CPB strength. Therefore, the reduction of C–S–H gels and the appearance of thaumasite led to a decrease in CPB strength.

### 3.4. Pore Structure Analysis

In this paper, Image J software was used to analyze the pore structure on the basis of the SEM images of the specimens with a curing age of 28 days and a magnification of 1000 times of the different groups. A pore structure analysis diagram is shown in Figure 10. The pore analysis results are shown in Table 3. The black area in the picture corresponds to the pore structure, and the white areas are mainly the generated hydration products. From Figure 10a corresponding to the SF0 group, it can be understood that except for a small area of pores on the left side of the picture, most of the pore areas were not clustered but mostly existed in the form of small pores. Part of the hydration products was attached near the pores, and some pores were wrapped by the hydration products. As the curing time increases, the generated Mg–Al type LDHs and C–S–H gel began to fill the pores continuously, thus compressing the pores, making the microstructure more compact and reducing the porosity. The SF5 group is shown in Figure 10b. It is clear that the red area was smaller than the SF0 group. Not only did we not observe the pores that forming the area like the SF0 group, but the particle size of these small pores was also observed to be significantly smaller than those in the SF0 group. These small pores were distributed near the hydration products, indicating that compared with the SF0 group, the hydration reaction of the mixture in the SF5 group occurs more readily, leading to an increase in Mg–Al type LDHs and C–S–H gel, and the degree of pore compression was higher than that in the SF0 group. During the UCS test, with increasing axial load, due to the small number of pores and the small pore volume, the load on the test specimen was uniform at levels approximately, so the specimen was subjected to more complete force during the entire uniaxial compression process and thus had a higher UCS value. From the test results, the porosity of the SF5 group was 18.38%, which was significantly smaller than that of the SF0 group by 23.08%. From the perspective of macro-mechanical properties, the UCS value of the SF5 group was greater than that of the SF0 group. The pore maps of the SF10, SF15, and SF20 groups are shown in Figure 10c–e, respectively. The porosity of the SF10, SF15, and SF20 groups was higher than that of the SF5 group. This may be due to the reduction of the C–S–H gels and the micro-cracks generated by thaumasite.

## 4. Conclusions

With increasing silica fume, the UCS tended to increase first and then decrease. When the amount of silica fume was approximately 5%, CPB with a larger UCS could be obtained.When the amount of silica fume increased from 0% to 5%, because silica fume had good activity and small particles, more C–S–H gels and Mg–Al type LDHs were produced in CPB, and it became increasingly denser, thus increasing the CPB strength.C–S–H gels were the main source of CPB strength. As the amount of silica fume gradually increased from 5% to 15%, thaumasite gradually produced inside the CPB, reducing the content of C–S–H gels. Due to the expansibility of thaumasite, a large number of micro-cracks were generated in the CPB, leading to a decrease in the CPB strength.

## Figures and Tables

**Figure 1 materials-13-00372-f001:**
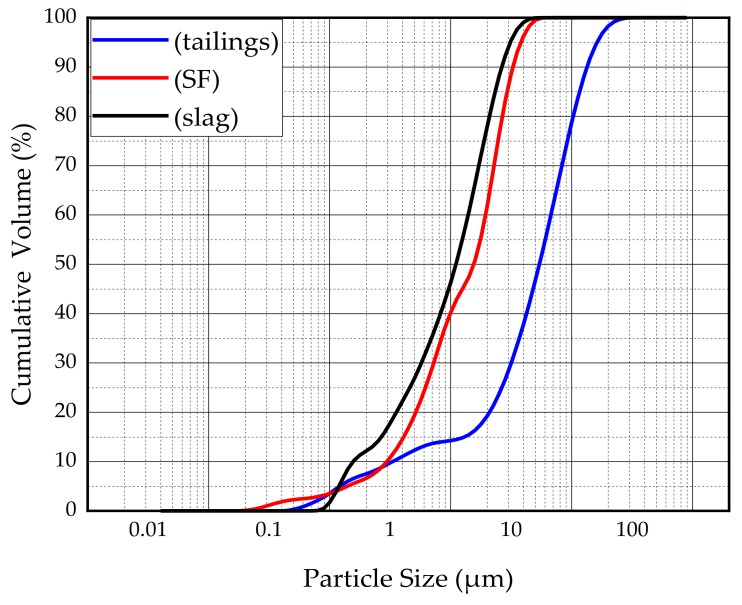
Particle size distribution of the tailings and binders.

**Figure 2 materials-13-00372-f002:**
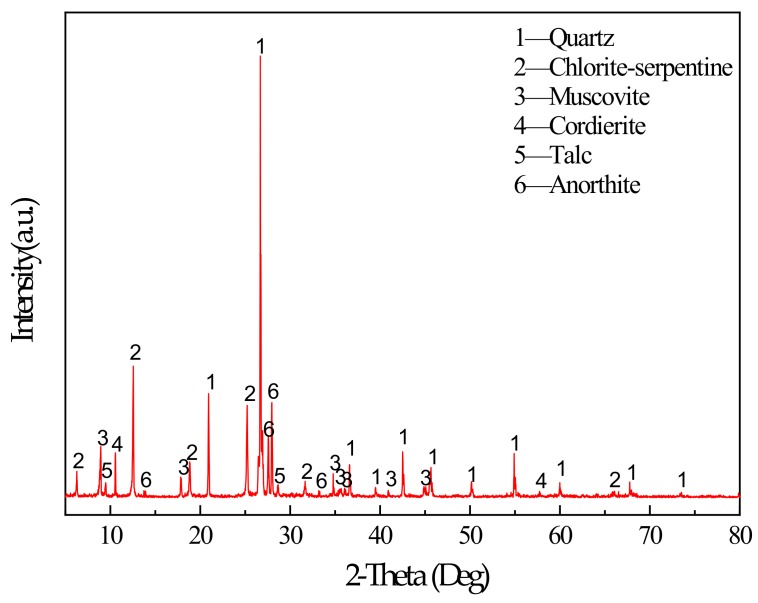
XRD pattern of tailings.

**Figure 3 materials-13-00372-f003:**
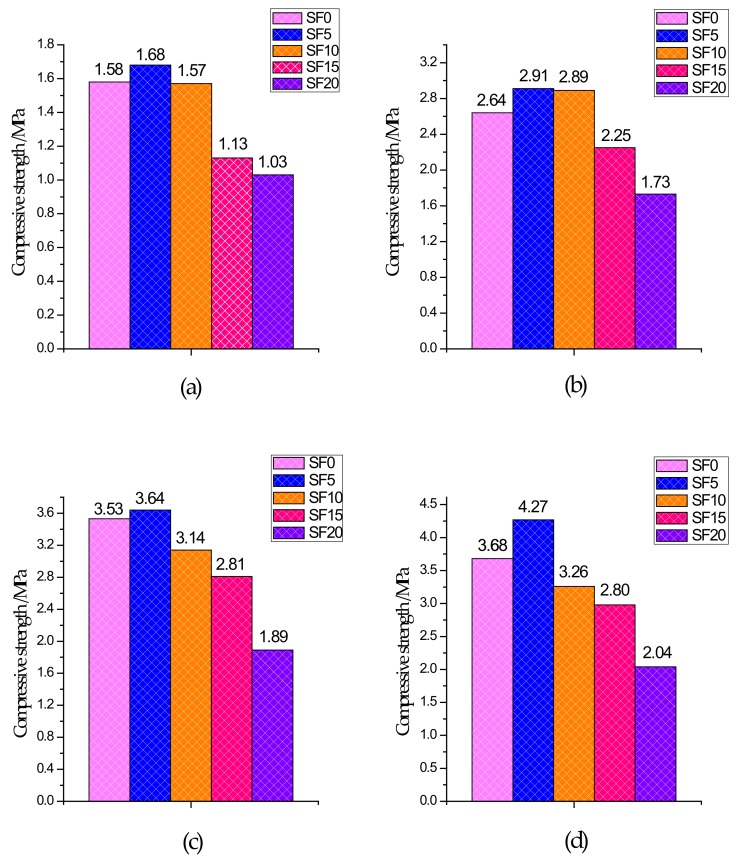
Compressive strength of different groups: (**a**) 3 days; (**b**) 7 days; (**c**) 14 days; and (**d**) 28 days.

**Figure 4 materials-13-00372-f004:**
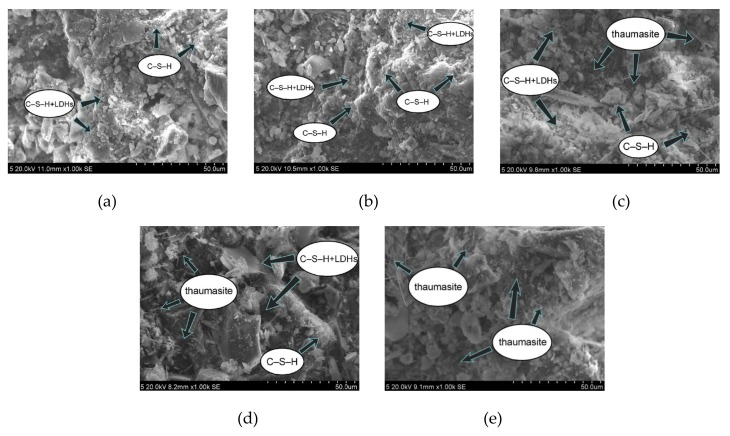
Micromorphology of cemented paste backfill (CPB) samples at 28 days: (**a**) SF0; (**b**) SF5; (**c**) SF10; (**d**) SF15; and (**e**) SF20.

**Figure 5 materials-13-00372-f005:**
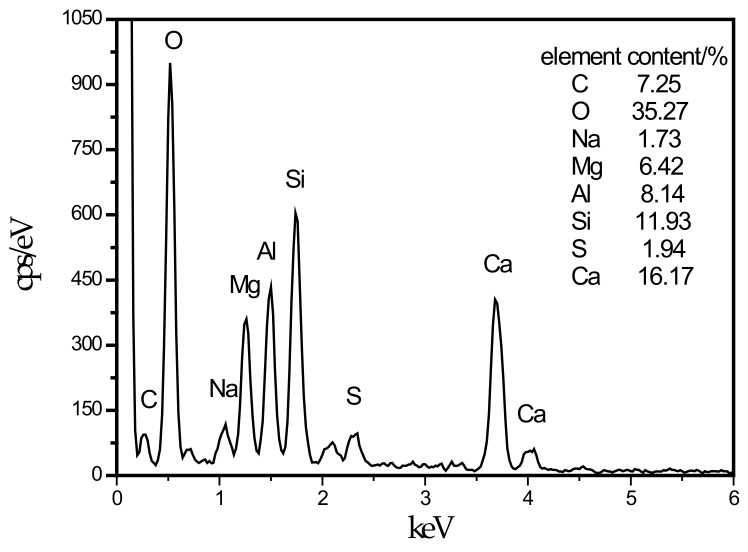
Energy dispersive spectrometry (EDS) energy spectrum of LDHs intimately intermix with the C–S–H.

**Figure 6 materials-13-00372-f006:**
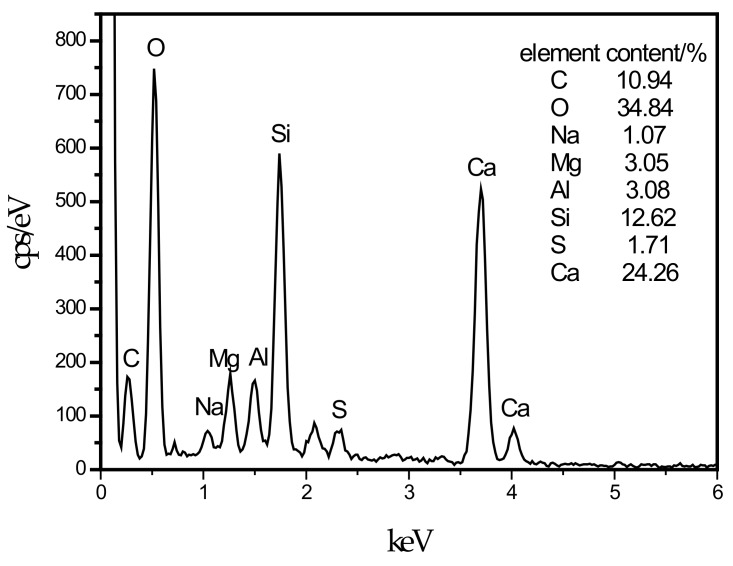
EDS energy spectrum of C–S–H.

**Figure 7 materials-13-00372-f007:**
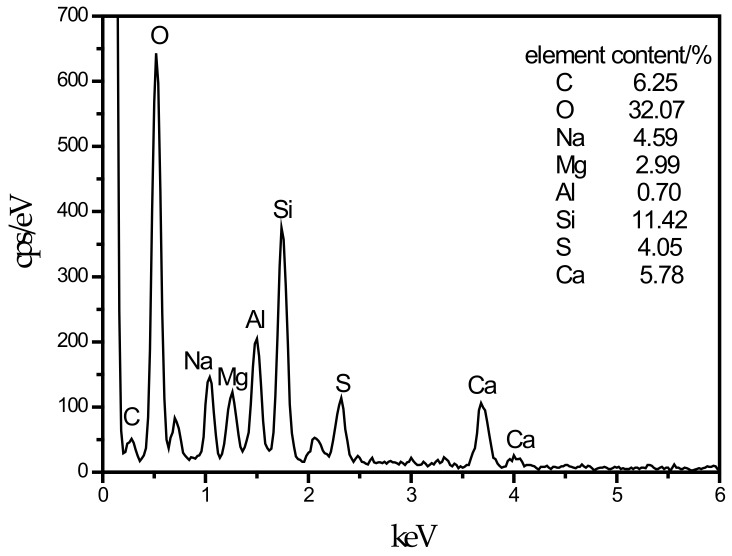
EDS energy spectrum of thaumasite.

**Figure 8 materials-13-00372-f008:**
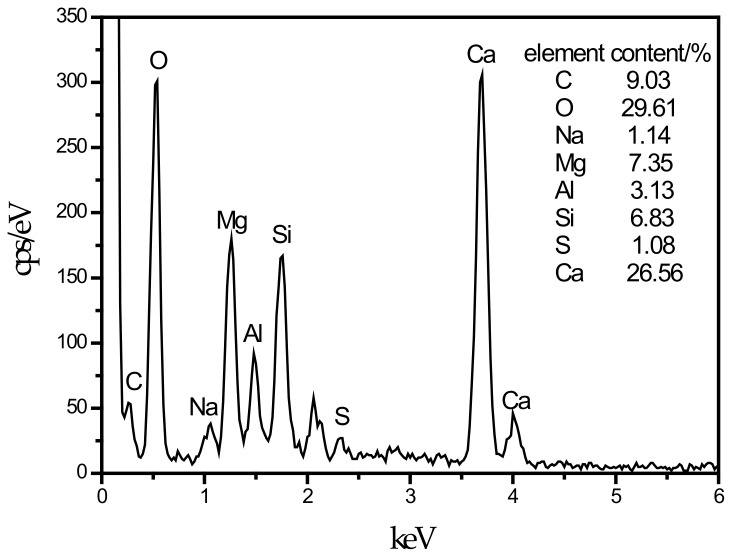
EDS energy spectrum of LDHs cover C–S–H.

**Figure 9 materials-13-00372-f009:**
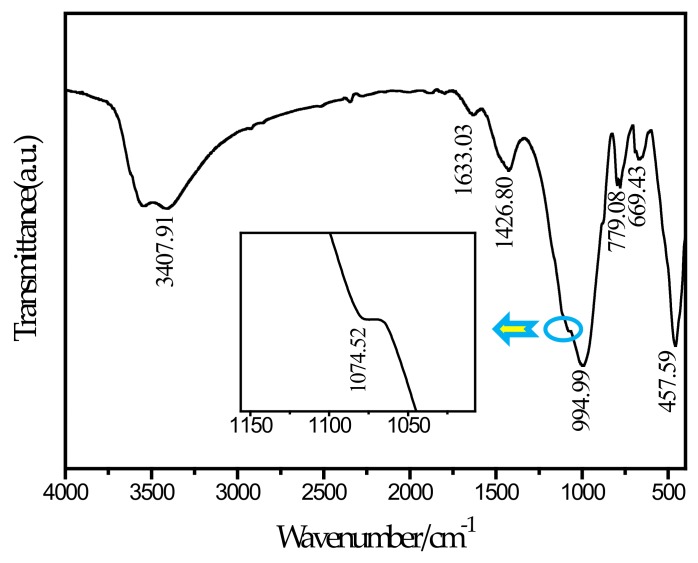
FTIR spectrum of SF20.

**Figure 10 materials-13-00372-f010:**
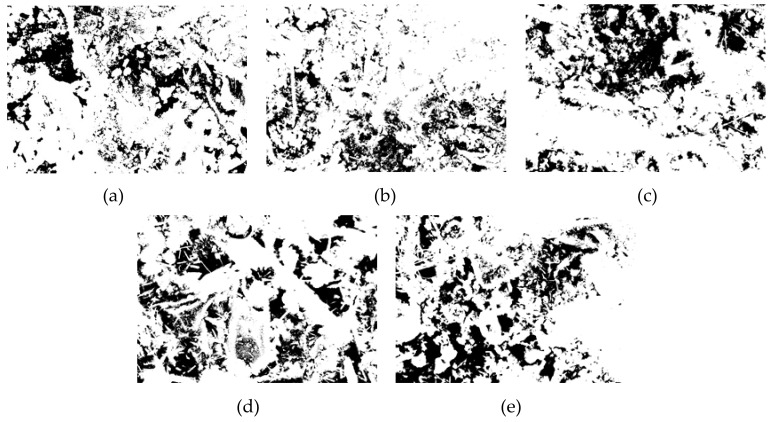
CPB pore structure analysis of different groups: (**a**) SF0; (**b**) SF5; (**c**) SF10; (**d**) SF15; and (**e**) SF20.

**Table 1 materials-13-00372-t001:** Chemical composition of raw materials (wt. %).

Materials	Cao	Fe_2_O_3_	SiO_2_	Al_2_O_3_	MgO	Na_2_O	K_2_O	SO_3_
Slag	40.75	0.55	27.51	16.20	7.77	0.26	0.25	2.91
SF	1.70	0.65	95.59	‒	0.22	‒	0.34	0.69
Tailings	3.06	35.49	40.62	10.81	6.60	‒	1.03	0.47

**Table 2 materials-13-00372-t002:** Details of the mixture proportions.

MIX ID	Binder Proportion (wt %)		Alkaline and Admixture (by Binder Mass %)		Water-Binder Ratio
	Slag	SF	Alkaline	Super Plasticizer	
SF0	100	0	4	0.2	2.1
SF5	95	5	4	0.4	2.1
SF10	90	10	4	0.6	2.1
SF15	85	15	4	0.6	2.1
SF20	80	20	4	1.0	2.1

**Table 3 materials-13-00372-t003:** Porosity of the CPB samples.

MIX ID	SF0	SF5	SF10	SF15	SF20
**Porosity %**	23.08	18.38	25.08	27.19	24.03

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
