# Peer review of "Preparation of a New Type of Cemented Paste Backfill with an Alkali-Activated Silica Fume and Slag Composite Binder"

_materials, 2020, doi:10.3390/ma13020372_

Round 1
Reviewer 1 Report
The paper presents an interesting and useful extension of use of alkali-activated technology as cementitious paste backfill. The paper need a minor correction before publish for public. My comments as following:
Page 1 Line 13
slag and silica fume (SF) as the binder, and tailings as the aggregate.
It is really confusing, The researcher mentioned in the line 12, cemented paste backfill then he mentioned using the tailing as aggregate. And No details about aggregate had mentioned in table 2, (the amount in Kg, or grading curve, Max aggregate size or is it fine or coarse aggregate???!!!!!) the unconfined compressive strength test has been done on paste or concrete or mortars? From values it seems paste, then in which test had the researcher used the tailing. ?
please make it clear
Page 1 Line 14
effects of five silica fume dosages of 0%, 5%, 10%, 15%, and 20% on the CPB performance were studied
Silica fume has been used as replacement with slag then it is not dosage !!!, you could write proportion of replacement.
Also at the end of the abstract (Line 24-26),, this is not clear explanation to understand the reason behind the reduction in strength with increase the SF replacement!!!????
Thaumasite ( Ca3Si(OH)6(CO3)(SO4)·12H2O )has also Ca and Si, you mentioned in page10 Line 279-280…increasing SF content decreased the Ca content which reduced the C-S-H gels, leading to a decrease in the CPB strength, and produced more Thaumasite which cause expansion and reduction in strength . Did you improve that in FTIR spectrum ??? or any other test?
Page 3 Line 109 and 110
Please add the producer place (Country? )
The main binder for this study was S95 fine slag powder produced by Shandong Kangjing New Material Technology Co., Ltd. (Country? ), and silica fume was produced by Luoyang Yumin Micro Silica Fume Co., Ltd (Country? )
Page 3 Line 127 and 128
water to binder ratio of 2.1 and an aggregate to binder ratio of 7
2.1 it is too much is it % ratio ? is it for paste , mortar or concrete ?
Page 4 Line 144
The mould containing the mixture was placed in a concrete standard curing box in which the temperature was controlled at (20±1) C, and the humidity was controlled at ≥96%.
Have you sealed the specimens ?? please put more details about curing
Page 4 Line 163-171
Under the action of the alkali stimulated catalyst, a large amount of C-S-H gels and C-A-S-H gels were generated, which became the main source of CPB strength. These generated gels compress the pores, while at the same time, because the silica fume particles are very small, they have a good filling effect on the pores and make the microstructure denser. Therefore, the UCS was higher in the SF5 group than in the SF0 group. With an increase in the amount of silica fume, UCS showed a trend of increasing first and then decreasing, which is similar to the experimental results of silica fume mixed with concrete presented by Wetzel et al.[37] and Alanazi et al.[38]. It is worth noting that for the SF15 and SF20 groups, the UCS value of all curing ages is significantly lower than that of SF0, which indicates that the more silica fume that is used, the easier it is to decrease the UCS.
Adding or replacing with silica fume producing high backing density, high pozzolanic reaction and forming more C-S-H and C-A-S-H, then increasing the strength. Why it caused this reduction in strength with replacing of 10% , 15% and 20%???? Please try to find good and reasonable explanation supported by XRD or TGA or FTIR test for this behaviour !!!
Page 6 Figure 2 is it for paste or concrete ???
Please add this detail for all figures
Figures 3 and 6 showed high MgO content,
Have you thought about forming another MgO phases? like hydrotalcite Mg6Al2CO3(OH)16·4(H2O)?
Figure 8 is similar to Figure 2.
Since you have used the Image J software to calculate the porosity%, my suggestion is to put instead the images from that analysis with black and white only without grey level, or remove Figure 8
Page 10 line 290
Table 4 instead of Results(%), please put Porosity %, is it for paste, mortar or concrete specimens??? please mention that in the table’s title.
Page 10 Line 297
Tip. 3, Please put more explanation behind the reduction in strength
Author Response
Response to Reviewer 1 Comments
Point 1: Page 1 Line 13
slag and silica fume (SF) as the binder, and tailings as the aggregate.
It is really confusing, the researcher mentioned in the line 12, cemented paste backfill then he mentioned using the tailing as aggregate. And No details about aggregate had mentioned in table 2, (the amount in Kg, or grading curve, Max aggregate size or is it fine or coarse aggregate???!!!!!) the unconfined compressive strength test has been done on paste or concrete or mortars? From values it seems paste, then in which test had the researcher used the tailing? Please make it clear
Response 1: Thank you for your valuable suggestion. The particle size distribution of tailings was shown in Figure 1 (Page 4). According to the Figure 1, tailings can be determined as fine aggregate. The maximum aggregate size was 398.2μm. The bulk density of the aggregate was 1752.2kg/m3. Changes in (Page 3 Line 125-129)
The unconfined compressive strength test and other tests have been done on cemented paste backfill (CPB).
Point 2: Page 1 Line 14
effects of five silica fume dosages of 0%, 5%, 10%, 15%, and 20% on the CPB performance were studied
Silica fume has been used as replacement with slag then it is not dosage !!!, you could write proportion of replacement.
Response 2: According to reviewer’s comment, “dosage” was changed into “proportion of replacement”. Change in (Page 1 Line13-15)
Point 3: Also at the end of the abstract (Line 24-26), this is not clear explanation to understand the reason behind the reduction in strength with increase the SF replacement!!!????
Response 3: Thanks for your comment, more languages are used to explain the reason behind the reduction in strength with increase the SF replacement. (Page 1 Line 24-27)
Point 4: Thaumasite (Ca3Si(OH)6(CO3)(SO4)·12H2O)has also Ca and Si, you mentioned in page10 Line 279-280…increasing SF content decreased the Ca content which reduced the C-S-H gels, leading to a decrease in the CPB strength, and produced more thaumasite which cause expansion and reduction in strength. Did you improve that in FTIR spectrum ??? or any other test?
Response 4: As reviewer’s comment, this problem was solved by a combination of SEM, EDS, and FTIR. (SEM-EDS Page8 Line234-241 and FTIR Page10 Line282-288) At the same time, we realized that the position of the section on page 10 line 279-280… was not appropriate. We turn this paragraph to lines 282-288 on page 10 for a more complete explanation.
Point 5: Page 3 Line 109 and 110
Please add the producer place (Country?) The main binder for this study was S95 fine slag powder produced by Shandong Kangjing New Material Technology Co., Ltd. (Country?), and silica fume was produced by Luoyang Yumin Micro Silica Fume Co., Ltd (Country?)
Response 5: It is our negligence and we are sorry about this. We have added the relevant information about the material manufacturer. (Page 3 Line 122,123)
Point 6: Page 3 Line 127 and 128
Water to binder ratio of 2.1 and an aggregate to binder ratio of 7; 2.1 it is too much is it % ratio ? is it for paste , mortar or concrete ?
Response 6: Cemented Paste Backfill (CPB) is a toothpaste-like paste and usually comprised of binder, solid waste materials such as tailings, coal gangue, fly ash, and water. Compared with concrete, CPB has large slump and low strength, but it only needs to meet the backfill requirements. Considering the cost of backfill, the content of binder is very low. Therefore, its water-binder ratio and aggregate-binder ratio are much larger than that of concrete. Water-binder ratio of 2.1 is normal for CPB. This value is not a percentage.
Point 7: Page 4 Line 144
The mould containing the mixture was placed in a concrete standard curing box in which the temperature was controlled at (20±1) oC, and the humidity was controlled at ≥96%. Have you sealed the specimens?? Please put more details about curing.
Response 7: Thank you for your valuable comment, we have added more details about curing conditions in the manuscript. (Page 5 Line 170-173)
Point 8: Page 4 Line 163-171
Under the action of the alkali stimulated catalyst, a large amount of C-S-H gels and C-A-S-H gels were generated, which became the main source of CPB strength. These generated gels compress the pores, while at the same time, because the silica fume particles are very small, they have a good filling effect on the pores and make the microstructure denser. Therefore, the UCS was higher in the SF5 group than in the SF0 group. With an increase in the amount of silica fume, UCS showed a trend of increasing first and then decreasing, which is similar to the experimental results of silica fume mixed with concrete presented by Wetzel et al.[37] and Alanazi et al.[38]. It is worth noting that for the SF15 and SF20 groups, the UCS value of all curing ages is significantly lower than that of SF0, which indicates that the more silica fume that is used, the easier it is to decrease the UCS.
Adding or replacing with silica fume producing high backing density, high pozzolanic reaction and forming more C-S-H and C-A-S-H, then increasing the strength. Why it caused this reduction in strength with replacing of 10%, 15% and 20%???? Please try to find good and reasonable explanation supported by XRD or TGA or FTIR test for this behaviour!!!
Response 8: As your coments, based on SEM-EDS, we have used FTIR to further prove this phenomenon. The explanation of reduction in strength with replacing of 10%, 15% and 20% were added in page 10 line 282-288. Thaumasite has been proven to be a swellable hydration product and easily cause micro-cracks in CPB [49]. With the increase of silica fume replacement proportion, the content of calcium element decrease, that is because the added silicon is not enough to produce more hydration products with the other elements, so the C-S-H gels generated during the same period are reduced. Meanwhile, C-S-H gels were the main source of CPB strength. Due to expansibility of thaumasite, a large number of micro-cracks were generated in the CPB, leading to a decrease in the CPB strength. Therefore, the reduction of C-S-H gels and the appearance of thaumasite led to a decrease in CPB strength. (Page 10 Line 282-288)
Point 9: Page 6 Figure 2 is it for paste or concrete ???
Please add this detail for all figures
Response 9: Figure 2 is for Cemented Paste Backfill (CPB). This detail for all figures were added. Because all the figures are for CPB, CPB is added to the title of Figures. (Page 7 Figure 4) Due to two figures were added, Figure 2 has been renumbered as Figure 4.
Point 10: Figures 3 and 6 showed high MgO content,
Have you thought about forming another MgO phases? Like hydrotalcite Mg6Al2CO3(OH)16·4(H2O)?
Response 10: Thank you for your valuable suggestion. We found through analysis that this is indeed the result of the hydrotalcite intertwine with C-S-H gels and cover on C-S-H gels which is similar to reference [43]. (Page 7 Line 206-211) In addition, irrelevant reaction equations were deleted.
Point 11: Figure 8 is similar to Figure 2.
Response 11: Figure 8 is for Cemented Paste Backfill (CPB). This detail for all figures were added in Figure 8. Because all the figures are for CPB, CPB is added to the title of Figure 8. (Page 11 Line 320 Figure 10) Due to two figures were added, Figure 8 has been renumbered as Figure 10.
Point 12: Since you have used the Image J software to calculate the porosity%, my suggestion is to put instead the images from that analysis with black and white only without grey level, or remove Figure 8
Response 12: According to reviewer’s comments, we put instead the images from that analysis with black and white only without grey level. Therefore, we changed the description of colors in the image. (Page 11 Line 293) Due to two figures were added, Figure 8 has been renumbered as Figure 10. (Page 11 Figure 10)
Point 13: Page 10 line 290
Table 4 instead of Results (%), please put Porosity %, is it for paste, mortar or concrete specimens??? please mention that in the table’s title.
Response 13: Thanks for your comment, we put “Porosity %” instead of “Results (%)”. It’s for CPB specimens, we mention it in the table’s title. (Page 11 Line 321)
Point 14: Page 10 Line 297
Tip. 3 Please put more explanation behind the reduction in strength
Response 14: According to your comments, we explained the reason for the decrease in intensity in more detail. (Page 12 Line 328-331)
Reviewer 2 Report
The present work deals with an interesting and current subject. The authors described a new type of cemented paste backfill (CBP) prepared using alkali activated binder based on the slag blended with silica fume and iron ore tailings as the aggregate. Backfilling of the voids created by mining provide an increased level of stability to the ore body. In addition, it is a suitable and economic way of utilization of mining wastes. In this work, the tailings used to prepare CBP material contained 35.5 wt. % Fe2O3. I wonder if these tailings can be considered as a source of iron in the nearest future.
The authors investigated the strength formation mechanism of CPB and determined the optimal dosage of silica fume to achieve the highest unconfined compressive strength. The conclusions are solid and adequately supported by experimental evidence. Overall, the paper seems to be in good shape. However, the following comments should be considered before publication:
The raw materials (slag, silica fume and tailings) should be characterized with respect to their specific surface area or particle size distribution. What was mineralogical composition of the tailings? Line 123 – The heading of Table 1 “Chemical properties of raw materials” should be corrected (e.g. “Chemical composition of…”). Since the content of all oxides in Table 1 is expressed in wt. %, this should be indicated in the heading of the table. Similarly, the content of the alkaline solution and the ratio of water to binder are the same for all formulations in Table 2. Thus, they should also be indicated in the heading of the table. The corresponding columns should be removed from Table 2. In Section 2.2 (Line 128) it is indicated that an alkali content in the mixture was 4% by binder mass. However, from Table 2 it follows that an alkaline solution content in the mixture was 4% by mass of the binder. Which content is correct? What was the ratio of Na2O in alkaline agent (NaOH) to the weight of the binder? Table 3 and Fig. 1 seem to present the same information regarding compressive strength. Authors should consider keeping one of them only. In Fig. 7 Transmittance scale unit should be indicated as a.u. (arbitrary units). Some papers relevant to this study have not been considered. The effect of silica fume on the properties of alkali activated slag was studied in:
Collins F, Sanjayan JG. Effects of ultra-fine materials on workability and strength of concrete containing alkali-activated slag as the binder. Cem Concr Res 1999;29:459–62. https://doi.org/10.1016/S0008-8846(98)00237-3
Rostami, K. Behfarnia, The effect of silica fume on durability of alkali activated slag concrete, Constr. Build. Mater. 134 (2017) 262–268. https://doi.org/10.1016/j.conbuildmat.2016.12.072
A.A. Ramezanianpour, M.A. Moeini, Mechanical and durability properties of alkali activated slag coating mortars containing nanosilica and silica fume, Constr. Build. Mater. 163 (2018) 611–621. https://doi.org/10.1016/j.conbuildmat.2017.12.062
A.M. Rashad, M.H. Khalil, A preliminary study of alkali-activated slag blended with silica fume under the effect of thermal loads and thermal shock cycles, Constr. Build. Mater. 40 (2013) 522–532. https://doi.org/10.1016/j.conbuildmat.2012.10.014.
It is recommended that authors expand the review section of the manuscript and discuss their results in comparison with the results of the mentioned works.
Author Response
Response to Reviewer 2 Comments
Point 1: The raw materials (slag, silica fume and tailings) should be characterized with respect to their specific surface area or particle size distribution.
Response 1: Thank you for your valuable suggestion. The particle size distribution of the raw materials (slag, silica fume and tailings) was shown in Figure 1.(Page 4) Their specific surface area were added on Page 3 in Line 123-124,128-129.
Point 2: What was mineralogical composition of the tailings?
Response 2: Thanks for your comment, X-ray diffraction (XRD) test was performed to test the mineralogical composition of tailings. Mineralogical composition of the tailings are Quartz, Chlorite-serpentine, Muscovite, and Anorthite, as shown in Figure 2 (Page 3 Line131-134.)
Point 3: Line 123 – The heading of Table 1 “Chemical properties of raw materials” should be corrected (e.g. “Chemical composition of…”). Since the content of all oxides in Table 1 is expressed in wt. %, this should be indicated in the heading of the table. Similarly, the content of the alkaline solution and the ratio of water to binder are the same for all formulations in Table 2. Thus, they should also be indicated in the heading of the table. In Section 2.2 (Line 128) it is indicated that an alkali content in the mixture was 4% by binder mass. However, from Table 2 it follows that an alkaline solution content in the mixture was 4% by mass of the binder. Which content is correct?
Response 3: Thanks for your comment, the heading of Table 1 “Chemical properties of raw materials” has been corrected to "Chemical composition of raw materials (wt %)" (Page 4 Line 148). (By binder mass %) was added in the first row of Table 2. (Page 5 Line 179) In Section 2.2 (Line 153-154), “an alkali content in the mixture was 4% by binder mass” is right. In table 2, "Alkaline solution" was corrected to "Alkaline (By binder mass%)" and another wrong word "Water to binder“was corrected to”Water-binder ratio". (Page 5 Table 2)
Point 4: What was the ratio of Na2O in alkaline agent (NaOH) to the weight of the binder?
Response 4: Thanks for your comment, the ratio of Na2O in alkaline agent to the weight of the binder is 3.1%. (Page 5 Line 153-154).
Point 5: Table 3 and Fig. 1 seem to present the same information regarding compressive strength. Authors should consider keeping one of them only.
Response 5: Thanks for your comment, Table 3 was deleted and the UCS values were add at the Figure 1. (Page 6 Figure 3) Due to two figures were added, Figure 1 has been renumbered as Figure 3. The slump test results in the original Table 3 were added on page 5 in line 166-167.
Point 6: In Fig. 7 Transmittance scale unit should be indicated as a.u. (arbitrary units).
Response 6: Thanks for your comment, transmittance scale unit was indicated as a.u. in Figure 7. (Page 10 Line 281) Due to two figures were added, Figure 7 has been renumbered as Figure 9.
Point 7:Some papers relevant to this study have not been considered. The effect of silica fume on the properties of alkali activated slag was studied in:
Collins F, Sanjayan JG. Effects of ultra-fine materials on workability and strength of concrete containing alkali-activated slag as the binder. Cem Concr Res 1999;29:459–62. https://doi.org/10.1016/S0008-8846(98)00237-3
Rostami, K. Behfarnia, The effect of silica fume on durability of alkali activated slag concrete, Constr. Build. Mater. 134 (2017) 262–268. https://doi.org/10.1016/j.conbuildmat.2016.12.072
A.A. Ramezanianpour, M.A. Moeini, Mechanical and durability properties of alkali activated slag coating mortars containing nanosilica and silica fume, Constr. Build. Mater. 163 (2018) 611–621. https://doi.org/10.1016/j.conbuildmat.2017.12.062
A.M. Rashad, M.H. Khalil, A preliminary study of alkali-activated slag blended with silica fume under the effect of thermal loads and thermal shock cycles, Constr. Build. Mater. 40 (2013) 522–532. https://doi.org/10.1016/j.conbuildmat.2012.10.014.
It is recommended that authors expand the review section of the manuscript and discuss their results in comparison with the results of the mentioned works.
Response 7: Thanks for your comments, these papers have been added into the introduction and been discussed from three aspects of concrete, motar, and paste (Page 2 Line 97-106, Page 3 Line 109-110). Although these papers have very important reference value, the concrete, mortar, and paste studied in these papers were not used for backfill mining like this paper.